# Interfacial and Foaming Properties of Tailor-Made Glycolipids—Influence of the Hydrophilic Head Group and Functional Groups in the Hydrophobic Tail

**DOI:** 10.3390/molecules25173797

**Published:** 2020-08-20

**Authors:** Rebecca Hollenbach, Annika Ricarda Völp, Ludwig Höfert, Jens Rudat, Katrin Ochsenreither, Norbert Willenbacher, Christoph Syldatk

**Affiliations:** 1Technical Biology, Institute of Process Engineering in Life Sciences II, Karlsruhe Institute of Technology, 76131 Karlsruhe, Germany; ludwighoefert@web.de (L.H.); jens.rudat@kit.edu (J.R.); katrin.ochsenreither@kit.edu (K.O.); christoph.syldatk@kit.edu (C.S.); 2Applied Mechanics, Institute of Mechanical Process Engineering and Mechanics, Karlsruhe Institute of Technology, 76131 Karlsruhe, Germany; annika.voelp@kit.edu (A.R.V.); norbert.willenbacher@kit.edu (N.W.)

**Keywords:** glycolipids, biosurfactants, structure–function relationship, interfacial tension, interfacial rheology, foam stability, foam rheology, bubble size distribution

## Abstract

Glycolipids are a class of biodegradable surfactants less harmful to the environment than petrochemically derived surfactants. Here we discuss interfacial properties, foam stability, characterized in terms of transient foam height, gas volume fraction and bubble diameter as well as texture of seven enzymatically synthesized surfactants for the first time. Glycolipids consisting of different head groups, namely glucose, sorbitol, glucuronic acid and sorbose, combined with different C10 acyl chains, namely decanoate, dec-9-enoate and 4-methyl-nonanoate are compared. Equilibrium interfacial tension values vary between 24.3 and 29.6 mN/m, critical micelle concentration varies between 0.7 and 3.0 mM. In both cases highest values were found for the surfactants with unsaturated or branched tail groups. Interfacial elasticity and viscosity, however, were significantly reduced in these cases. Head and tail group both affect foam stability. Foams from glycolipids with sorbose and glucuronic acid derived head groups showed higher stability than those from surfactants with glucose head group, sorbitol provided lowest foam stability. We attribute this to different head group hydration also showing up in the time to reach equilibrium interfacial adsorption. Unsaturated tail groups reduced whereas branching enhanced foam stability compared to the systems with linear, saturated tail. Moreover, the tail group strongly influences foam texture. Glycolipids with unsaturated tail groups produced foams quickly collapsing even at smallest shear loads, whereas the branched tail group yielded a higher modulus than the linear tails. Normalized shear moduli for the systems with different head groups varied in a narrow range, with the highest value found for decylglucuronate.

## 1. Introduction

Foams are thermodynamically unstable systems of bubbles dispersed in a solution stabilized by surfactants which can have a wide range of possible applications, e.g., in agriculture, cosmetics, food, fire-fighting, oil recovery and wastewater treatment [1,2]. Foams become destabilized by drainage, coarsening and coalescence [1,3,4]. These destabilizing mechanisms are related to dynamic interfacial tension, interfacial elasticity and interfacial viscosity [5,6,7,8]. Therefore, foam characteristics, such as foam stability, coarsening rates and bubble size distribution, strongly depend on the surfactant used for stabilization. Surfactants are amphiphilic molecules of a hydrophilic and a hydrophobic moiety. Therefore, they adsorb at interfaces and stabilize them by lowering interfacial tension as well as providing a barrier against aggregation and coalescence. Certain surfactants may also induce interfacial viscoelasticity.

Glycolipids are a class of surfactants consisting of a lipid moiety linked to a carbohydrate. They are produced chemically or biotechnologically, either by fermentation or by enzymatic synthesis. Biotechnological production has the advantage over chemical production of milder reaction conditions, as neither high temperatures nor toxic catalysts nor protection and deprotection steps are required [9]. The structural diversity of the fermentatively produced glycolipids is limited by the metabolism of the microorganisms, whereas in enzymatic synthesis there are theoretically no limits to this diversity [9]. Hence, enzymatic synthesis is a promising strategy for production of tailor-made glycolipids. Knowledge of the structure–function relationship is essential to select suitable head and tail groups for the respective application. While numerous studies dealt with the foaming properties of petrochemically derived surfactants, only few studies are available on glycolipid applications in foams, although glycolipids are biodegradable and less harmful to the environment than petrochemical surfactants, qualifying them as sustainable alternatives [10,11,12,13,14,15]. Another advantage of glycolipids over conventional surfactants is the temperature-insensitivity of their physicochemical properties which allows for applications over a broad temperature range [16,17]. Glycolipids are non-toxic and skin friendly, and some of them exhibit antimicrobial activity [10,11,18,19]. Hence, glycolipids are highly interesting for applications in food, cosmetics and pharmaceutics [20,21,22].

For alkyl glycosides, it is known that foam stability of molecules with a monosaccharide as head group is best with a C10 tail group, while shorter and longer chain length decreases foam stability [23,24]. However, alkyl glycosides have a higher skin irritation potential than glycolipids with an ester bond instead of the ether bond [19]. Thus, sugar acylates have a broader range of applications. Zhang et al. reported that acylated monosaccharides with a C10 fatty acid chain exhibit higher foaminess, higher foam stability and lower critical micelle concentration (CMC) than laurates [25]. Therefore, glycolipids with an ester bond and tail groups of 10 carbons were enzymatically synthesized in this study.

Head groups are reported to have only minor effects on interfacial tension and CMC [24,26], but alkyl glucosides with different head groups showed differences in foam stability [24]. However, the knowledge of the influence of head groups on foaming properties is limited to foam stability tests in a time range of 5 min.

Foaming properties are related to interfacial properties like dynamic interfacial tension and interfacial rheology [1,3,4,5,6,7,8,27]. Foam rheology has been shown not only to depend on Laplace pressure within the bubbles and gas volume fraction but also on interfacial elasticity [28]. Interfacial elasticity and interfacial viscosity also have an impact on foam stability. A higher interfacial viscosity results in foams with higher resistance against coarsening and foam rupture leading to slower foam decay [4,7,8]. However, only the study of Razafindralambro et al. addresses interfacial rheology of two glycolipids with different head groups, i.e., glucose octanoate and octyl glucuronate [27].

To the best of our knowledge, a comparative study on interfacial and foaming properties of different glycolipids focusing on acylated monosaccharides with a specific tail length has not been described in the literature yet. In this study, seven glycolipids with C6 head groups and C10 tail groups were enzymatically synthesized and evaluated (Figure 1). Four different sugar(-derivatives) were applied as a head group, namely the aldose glucose, the ketose sorbose, the uronic acid glucuronic acid and the alditol sorbitol. Fatty acid tails were either saturated, unsaturated or branched. In this study, foaming properties, including foam decay, transient gas volume fraction and mean bubble diameter as well as foam rheology, were related to interfacial dilatational viscoelasticity and glycolipid structure.

## 2. Results

Tailor-made glycolipids synthesized and investigated in this study had a purity of at least 95%. CMC was determined, and further measurements were conducted at glycolipid concentrations of twice the CMC. Dynamic interfacial tension, as well as interfacial elasticity and interfacial viscosity were determined as characteristic interfacial properties. Foam stability, i.e., transient foam height, bubble diameter and gas volume fraction, as well as foam shear modulus were analyzed for evaluation of the structure–function relationship of the different glycolipids.

### 2.1. Critical Micelle Concentration (CMC) and Dynamic Interfacial Tension

Critical micelle concentrations were determined for all glycolipids by Du Noüy ring method, and the results are summarized in Table 1. In general, glycolipids with a saturated fatty acid tail had significantly lower CMC values than those with monounsaturated fatty acid tails. The lowest CMC was determined for sorbitol monodecanoate (0.74 mM).

Dynamic interfacial tension measurements were performed using the pendant drop method in order to characterize the adsorption velocity of the glycolipids, which is known to have an impact on foaming properties. The equilibrium interfacial tension of the investigated glycolipids was in the range between 24 mN/m and 29 mN/m with the lowest interfacial tension of 24.3 ± 0.6 mN/m for decylglucuronate and the highest interfacial tension of 29.6 ± 1.9 mN/m for glucose mono-4-methylnonanoate (Table 1). Highest interfacial tension values were found for glycolipids with unsaturated or branched tail groups.

Decylglucuronate and glucose mono-4-methylnonanoate adsorbed much faster at the interface than the other investigated glycolipids, resulting in a faster reduction of the interfacial tension (Figure 2). Sorbitol monodecanoate had a significantly longer adsorption time than the other glycolipids (Figure 2a). Remarkably, the adsorption times of the glycolipids with a monounsaturated fatty acid tail were 2–4 times shorter than those of the corresponding saturated glycolipids (Figure 2b).

### 2.2. Interfacial Rheology

Interfacial rheological properties, i.e., interfacial elasticity and interfacial viscosity are known to influence foam properties since they contribute to resistance against coarsening, coalescence and drainage [5,6,7,8].

Interfacial dilatational elasticity of the investigated glycolipids was significantly different depending on the structure of the glycolipid (Figure 3a). The interfacial elastic moduli of solutions of glycolipids with branched or monounsaturated fatty acid tail exhibited no frequency dependence due to their fast adsorption kinetics. The molecular exchanges between interface and bulk during compression and dilation were fast enough to compensate glycolipid concentration gradients at the interface which leads to low apparent interfacial elasticities. The interfacial elastic moduli of glycolipids with linear fatty acid tail were higher and increase monotonically with increasing frequency. Diffusion of these glycolipids from the interface into the bulk and vice versa was slower than with branched or monounsaturated fatty acid tail but still present at these dilatational frequencies. These surfactant concentration fluctuations decrease with increasing dilatational frequency and cause the frequency dependence of the measured elastic moduli, which were thus not generated by intermolecular forces between a constant amount of surfactant molecules here.

Similar findings were observed for the dilatational interfacial viscosity (Figure 3b). Dynamic interfacial viscosity at 0.05 Hz was significantly higher for sorbose monodecanoate than for the other glycolipids. No significant differences were observed for glucose monodecanoate, sorbitol monodecanoate and decylglucuronate. Unsaturation in the fatty acid tail significantly lowered interfacial viscosity and branching resulted in the significantly lowest interfacial viscosity among the compared glycolipids. At frequencies of 0.1 Hz and higher, differences in the interfacial viscosity depending on the head groups were no longer significant while the glycolipids with unsaturated or branched hydrophobic tails had significantly lower interfacial viscosity over all frequencies tested. The tested glycolipids reached their equilibrium state of dilatational viscosity at the highest frequency tested.

### 2.3. Foam Stability

Foam stability is a key parameter when selecting suitable surfactants for the design of foamed commercial products [29].

The different investigated glycolipids were compared with regard to their ability to stabilize foam at glycolipid concentrations of twice the CMC value. Foams stabilized by glycolipids with a monounsaturated fatty acid tail were significantly less stable than those stabilized by glycolipids with saturated fatty acid tails (Figure 4). The decay time until reaching half of the initial foam height was 28 min for glucose monodec-9-enoate compared to 60 min for glucose monodecanoate and 25 min for sorbitol monodec-9-enoate compared to 30 min for sorbitol monodecanoate.

In contrast, branched fatty acid tails significantly increased foam stability: the decay time until 75% of the initial foam height was reached was 30 min for glucose mono-4-methylnonanoate compared to 4 min for glucose monodecanoate.

Foams stabilized by glucose mono-4-methyldecanoate, decylglucuronate and sorbose monodecanoate were most stable. Additionally, foam stability was higher with glucose monodecanoate than with sorbitol monodecanoate.

### 2.4. Bubble Size Distribution

Bubble size distribution was determined endoscopically. The initial bubble size distribution of foams stabilized by the seven glycolipids was quite similar except for decylglucuronate (Appendix B
Figure A1).

After 600 s, a bimodal distribution was observed for foams stabilized by glycolipids with unsaturated fatty acid tails due to the formation of some huge bubbles (Figure 5). In contrast, monomodal distribution was detected directly after foam formation, as well as at a foam age of 10 min for foams stabilized by glycolipids including saturated fatty acid moieties.

Remarkably, the initial Sauter bubble diameter of the decylglucuronate foam was higher than for the other saturated glycolipids while the growth rates of the bubbles were similar (Figure 6).

No differences in the Sauter bubble diameter of foams stabilized by glucose monodecanoate, sorbitol monodecanoate and sorbose decanoate foams were observed. Similar bubble size values were also found for the foam made from the glycolipid with the branched fatty acid moiety, glucose 4-methylnonanoate.

During the first 150 s of foam age, the Sauter bubble diameter of foams stabilized by saturated and unsaturated glycolipids did not differ, but did increase at higher foam ages faster for the foams of unsaturated glycolipids in contrast to those of saturated glycolipids. The coarsening rates of glucose monodecanoate, decylglucuronate, sorbitol monodecanoate and sorbose monodecanoate were in a range of 90–200 µm^2^/s (Appendix A). Branching had no significant effect on the coarsening rate, while unsaturated hydrophobic tails led to significantly higher coarsening rates of 2100 µm^2^/s for glucose monodec-9-enoate and 3400 µm^2^/s for sorbitol monodec-9-enoate.

### 2.5. Foam Gas Volume Fraction

The time evolution of gas volume fraction was determined using conductivity measurements. Although the initial gas volume fraction varies in a wide range (0.74–0.85), all foams reach essentially the same gas volume fraction of about 0.9 within 10 min. Remarkably, initial gas volume fraction of foams stabilized with decylglucuronate was significantly higher compared to the other glycolipids (Figure 7a). Branching in the fatty acid tail resulted in a lower value and slower raise of the gas volume fraction compared to non-branched glycolipids (Figure 7a). For foams made of sorbitol monodecanoate and glucose monodecanoate solutions, gas volume fraction increased faster than for the corresponding unsaturated glycolipids (Figure 7b).

### 2.6. Foam Elasticity

Foam elasticity characterizes the resistance of a foam against deformation. This is important for technological treatment during the production and transport of industrial foam products, for the texture and haptic sensation of food or cosmetic products.

Foam elasticity was characterized in terms of the shear modulus *G*_0_, normalized by the bubbles’ Laplace pressure given as the ratio of surface tension to Sauter bubble radius, at a gas volume fractions *φ* of 0.88 (Figure 8). Foams stabilized with unsaturated glycolipids could not be measured at gas volume fraction of 0.88 as foams were too fragile and collapsed under shear. The normalized shear modulus of foams stabilized with glucose mono-4-methyldecanoate was significantly higher than for glucose monodecanoate. Foam made of decylglucuronate solution exhibited a significantly higher normalized shear modulus compared to the other head groups. However, differences in foam elasticity between sorbitol monodecanoate and glycolipids containing another head group were not statistically significant.

## 3. Discussion

In this study, seven glycolipids were enzymatically synthesized in order to determine the influence of different head and tail groups on interfacial and foam characteristics (Figure 9).

With regard to the tail group, results of the direct comparison of two saturated glycolipids, i.e., glucose and sorbitol monodecanoate, and their unsaturated derivatives, glucose and sorbitol mondec-9-enoate, indicate a strong influence of unsaturation in the hydrophobic fatty acid tail on interfacial and foaming properties. The unsaturated glycolipids had a CMC about two times higher than those with a saturated fatty acid tail. These results are in good accordance with those for unsaturated fatty acids, potassium fatty acids and sodium 10-undecenoate for which it was reported that each unsaturation doubles CMC [30,31]. In the case of sophoroselipids and rhamnolipids, unsaturated fatty acid tails enhance the CMC value in contrast to saturated fatty acid tails [32,33].

Unsaturated glycolipids showed faster foam decay compared to saturated ones. Drainage is reported as the first instability effect to occur in foams [3]. Foam aging can be classified into three different stages by analysis of the ratio of change in foam volume to change of drained volume of the surfactant solution over time [1]. As long as this ratio is 1, foam volume decay is only drainage driven and no gas escapes from the foam. A transition to the second stage occurs when the ratio becomes greater than 1, because then additionally bubbles rupture and gas loss decreases foam volume. In the third stage, drainage has come to a halt and foam volume decay occurs due to bubble rupture only. For foams of glycolipids with unsaturated tail group, the classification of foam aging clearly indicates that foam decay of glucose monodec-9-enoate has a significantly shorter drainage-controlled stage 1 than glucose monodecanoate (Figure A2a,b), even though the resistance of glucose monodec-9-enoate stabilized foam against drainage is higher (Figure 6b). The subordinate role of drainage for the foam instability mechanisms of glycolipids with unsaturation is confirmed by higher coarsening rates compared to saturated glycolipids and by the bimodality of the bubble size distribution at a foam age of 10 min.

Interfacial elasticity, and to an even greater extent, interfacial viscosity, have already been reported to have stabilizing effects on foams [5,7,8]. Since interfacial elasticity and interfacial viscosity of unsaturated glycolipids were also lowest, their interfacial rheological properties are likely a reason for the lower stability of corresponding foams. The lower interfacial elasticity and interfacial viscosity may cause the bimodality of the bubble size distribution of the unsaturated glycolipids and not only coarsening but also coalescence occurred in these systems. Due to the unsaturation, van der Waals interactions between the tails are weaker than for the saturated glycolipids and this may be related to the higher CMC, lower interfacial elasticity and decreased foam stability.

Glucose mono-4-methylnonanoate was synthesized and investigated to study the influence of branching. The branched fatty acid tail led to a higher equilibrium interfacial tension compared to glucose *n*-monodecanoate while CMC values were similar. For lipopeptides, fatty acid tails containing iso-fatty acids have been reported to enhance biosurfactant surface activity and also for hydrocarbon surfactants with branched fatty acid tails stronger reduction of the interfacial tension were published [34,35,36]. However, similar equilibrium interfacial tension and CMC values have been reported for branched and non-branched tridecanyl maltoside and octyl glucoside [23,37].

Glucose mono-4-methylnonanoate stabilized foams exhibited superior stability compared to non-branched glycolipids. Interfacial elasticity and interfacial viscosity of glucose mono-4-methylnonanoate were significantly lower than those of glucose monodecanoate. Therefore, interfacial rheological parameters can be excluded as reasons for the higher foam stability of the branched chain glycolipid, as well.

In the literature, contradictory results were reported on foaming properties of branched glycolipids: while Koeltow et al. described a branched tridecanyl maltoside having higher foaminess and foam stability than an *n*-tridecanyl maltoside [23], Waltermo et al. reported lower foam stability for branched octyl glucoside compared to a *n*-octyl glucoside [37]. As the results of this study showed higher foam stability for the branched glucose monodecanoate compared to the unbranched glycolipid, it can be assumed that a minimal tail length is important for branching to enhance foam stability.

Drainage is slower in glucose mono-4-methylnonanoate stabilized foams as initial gas volume fraction was lower and a gas volume fraction of 0.9 was reached later compared to the non-branched glycolipids. Therefore, the retarded drainage is likely to be a reason for the higher foam stability with the branched glycolipid. This is supported by the classification of foam aging suggested by Lunkenheimer et al. [1]. Branching in the fatty acid chain leads to an extended drainage-controlled stage 1 (Figure A2a). Hence, for glucose mono-4-methylnonanoate, drainage is likely the dominating instability mechanism.

To investigate the influence of head groups, four glycolipids with different head groups were synthesized. Determined CMC values were between 0.7 mM to 1.5 mM.

With regard to interfacial properties, the CMC value for decylglucuronate was lower than that of glucose decanoate and also the adsorption of decylglucuronate at the interface was faster compared to glucose decanoate. This is likely due to the more hydrophobic character of decylglucuronate as the hydrophobic chain is not interrupted by a carbonyl group. Similar results were also observed for octylglucuronate compared to glucose octanoate; however, differences in the interfacial tension and interfacial rheology were reported [27]. Contrarily, we did not observe significant differences in interfacial tension or in interfacial rheology for glucose monodecanoate and decylglucuronate. The differences between the results reported by Razafindralambro et al. [27] and the results of this study might be due to the longer tail length of decylglucuronate and glucose monodecanoate, and therefore, the differences in the hydrophobic character of the molecules might be smaller and thus have less effect on interfacial properties.

Ducret et al. investigated CMC of glucose and sorbitol esters. For caprylates the CMC of the sorbitol ester was lower while for laurates the CMC for glucose esters was lower [38]. In this study, the CMC of sorbitol decanoate was lower than that of glucose decanoate and therefore it can be assumed that for tail length up to C10 sorbitol esters have lower CMC values than glucose esters as the hydrophilic lipophilic balance decreases with increasing tail length. The measured values for the decanoates are in-between the values for caprylates and laurates [38].

Concerning foam characteristics, decylglucuronate and sorbose monodecanoate stabilized foams exhibited superior stability compared to the foams made from the glycolipids with the other head groups. However, interfacial elasticity and interfacial viscosity of decylglucuronate and sorbose monodecanoate were similar to those of glucose and sorbitol monodecanoate, and accordingly the differences in foam stability of these glycolipids cannot be explained by their interfacial rheological properties.

Comparison of the foam stability found for the different head groups of this study with literature values shows that the investigated glycolipids have a comparatively high potential for foam stabilization. While for the investigated glycolipids foam half-life is at least 30 min, a foam half-life of less than 10 min is reported for rhamnolipids at a concentration of 10 times CMC [39]. For the biosurfactant surfactin, a residual foam volume after 20 min of 34% was published at a concentration of 5 times the CMC [40,41]. The synthetic surfactants methylestersulfonates (alkylchain length of 14–18 carbons) and polyoxyethylated dodecyl alcohol (3–9 ethoxy groups) show a half-life of no more than 3 min and 1.5 min at concentrations of 0.2 up to 5 times the CMC [2]. However, the comparability of foaming experiments between different laboratories is limited, as the results can vary considerably with different methods and different gases used for foaming.

Comparing the effect of the different head groups on foam decay revealed that for the most stable foams with decylglucuronate and sorbose monodecanoate drainage was the mechanism controlling foam decay (Figure A2a). The less stable foam made from sorbitol monodecanoate showed a shorter stage 1 than that made from glucose monodecanoate.

Although interfacial rheological properties alone do not explain foam stability and resistance against drainage sufficiently, foam stability as characterized by the foam height at 60 min foam age normalized to the initial foam height, correlated with the adsorption time required to reach equilibrium interfacial tension (Figure 10). Glycolipids characterized by shorter adsorption times exhibited higher foam stability. The glycolipids with a shorter adsorption time reach the interface faster and therefore stabilize the bubbles more efficiently. This is supported by the findings of Petkova et al., who determined a correlation between dynamic interfacial tension and foaminess for non-ionic surfactants [42].

In general, interfaces in foams of non-ionic surfactants are predominantly stabilized by repulsion forces between surfactant molecules [42]. In the case of glycolipids electrostatic repulsion contributes to repulsion forces due to the hydration of the head group [16,17,43,44]. Aldoses and ketoses exhibit different degrees of hydration [45]. Glucose is an aldose while sorbose is a ketose, glucuronic acid an uronic acid and sorbitol an alditol. Therefore, despite similar interfacial rheology the differences in dynamic interfacial tension and foaming properties between glucose monodecanoate, sorbose monodecanoate and sorbitol monodecanoate are likely due to their different hydration which causes differences in the repulsion forces between the surfactant molecules and consequently also in the foam films. However, the interactions between sugar head groups at interfaces are not well understood yet.

In summary, all investigated glycolipids exhibited promising foam stability compared to different synthetic surfactants as well as biosurfactants described in the literature [2,39,40,41]. Nevertheless, the results of this study indicate that ketoses are more suitable head groups for glycolipids than aldoses or alditols with respect to foam stabilizing properties. Furthermore, our results suggest a preference of branched fatty acid groups over unbranched or unsaturated fatty acid groups for foam applications. Sorbose monodecanoate yields the highest potential among the investigated glycolipids for application as foaming agent, because its foam performs best with respect to volume stability over time, rate of bubble size and gas volume fraction change. It finally also provides a high foam elasticity at a relatively low surfactant concentration of 0.2%.

## 4. Materials and Methods

### 4.1. Materials

Lipase B from *Candida antarctica*, immobilized on acrylic resin (iCalB) was purchased from Strem Chemicals (Strem chemicals Europe, Kehl, Germany). Vinyl decanoic acid and dec-9-enoic acid were acquired from Tokyo Chemical Industry Co., Ltd. (TCIEurope, Eschborn, Germany). 4-methyl nonanoic acid and glucuronic acid were purchased from VWR (Radnor, PA, USA). Glucose, sorbitol and all solvents (in HPLC grade) were acquired from Carl-Roth (Karlsruhe, Germany). Sorbose was a kind gift from Givaudan (Paris, France). 6-Decanoyl-d-glucose was purchased from Sohena (Tübingen, Germany).

### 4.2. Synthesis of Glycolipids

Substrates were mixed in equimolar ratio, 0.5 M sugar(derivate) and 0.5 M (vinyl-)fatty acid, and 10mg/mL iCalB in a 250 mL round bottom flask in 100 mL acetone. iCalB prefers primary hydroxyl groups and therefore esterification takes place at the primary hydroxyl group of the sugar(derivative) [46,47,48]. The samples were shaken at 50 °C and 600 rpm in a Laborota 4000 rotatory evaporator (Heidolph, Schwabach, Germany) at atmospheric pressure for 48 h. For the synthesis of glucose monodecanoate glucose and vinyldecanoate were used as substrates, for glucose monodec-9-enoate glucose and 9-decenoic acid, for glucose mono-4-methyl-nonanoate glucose and 4-methyl-nonanoate, for sorbitol monodecanoate sorbitol and vinyldecanoate, for sorbitol monodec-9-enoate sorbitol and 9-decenoic acid, for decylglucuronate glucuronic acid and decanol and for sorbose monodecanoate sorbose and vinyldecanoate.

### 4.3. Purification of Glycolipids

The obtained glycolipids were filtrated with a Büchner funnel and the filtrate was washed three times with ethyl acetate. The glycolipid containing solvent was evaporated with a rotatory evaporator at 40 °C and 240 mbar. Solids were subsequently purified by flash chromatography using a Reveleris Prep system from Büchi Labortechnik GmBH (Essen, Germany) and a Flash Pure Silica column (40 g, 53–80 Å). Mobile phase was made of chloroform (A) and methanol (B). A gradient was used for separation of products and residual substrates: starting from 100% A, a linear gradient was applied to 96% A and 4% B within 2 min. This ratio was held for 9 min, followed by another linear gradient to 90% A and 10% B in 2 min. This ratio was held for 6 min. Afterwards a linear gradient to 75% A and 25% B in 2 min was applied, and this ratio was held for 4 min, followed by a linear gradient to 100% B in 2 min, and this was held for 6 min. Peaks were collected and fractions controlled by TLC. Therefore, 5 µL of samples were spotted on Alugram Xtra SIL G plate from Machery-Nagel (Düren, Germany). For elution a mobile phase of chloroform: methanol: acetic acid was used (65:15:2, by vol). Compounds were visualized by anis aldehyde dying (anis aldehyde: sulfuric acid: acetic acid 0.5:1:100, by vol). Product containing fractions were collected and solvents were again evaporated with a S-Concentrator BaVC-300H from Helmut Saur Laborbedarf (Reutlingen, Germany). The purity of the products was checked by HPLC-ELSD.

### 4.4. HPLC-ELSD

HPLC analysis was performed according to Hollenbach et al., using a Kinetex EVO C18 (2.6 µm, 250 × 4.6 mm) from Phenomenex (Aschaffenburg, Germany) with an accompanying guard column (4 x 3.0 mm ID) of the same phase using an Agilent (Waldbronn, Germany) 1260 series liquid chromatograph equipped with a quaternary pump, an autosampler and a column oven [49]. For detection, an evaporative light scattering detector from BÜCHI Labortechnik (Essen, Germany) was used. Mobile phase was a gradient of acetonitrile (A) and water (B) with a total flow rate of 1mL/min. This method reliably separates monoesters from substrates and by-products such as diesters [49]. Only products with a purity of at least 95% determined by the area % of the HPLC chromatograms were used for further investigations.

### 4.5. Determination of Interfacial Tension

Interfacial tension was determined with a Lauda Tensiometer TD1 (Lauda-Königshofen, Germany) by the Du Noüy-ring method. Before the measurement, the tensiometer needed to be prepared by calibrating with a 500 mg calibration weight. A test vessel was filled with at least 2 mL glycolipid solution and placed on the stage of the tensiometer. A Du Noüy ring (19.1 mm diameter) was submerged at least 2-3 mm below the solution surface. After an equilibration time of 15 min, the measurement was started by lowering the stage manually. The maximum normal force before the lamella formed between ring and solution breaks is the uncorrected interfacial tension *σ_unc_*. The absolute interfacial tension *σ_abs_* is obtained by multiplying *σ_unc_* with a correction factor *f*. The correction factor f for the used ring was calculated as follows [50,51]:(1)$$f\;=\;0.8759\;+\;\frac{0.0009188}{\rho },$$
where *ρ* is the density of the test liquid.

The surface excess concentration *Γ* and the molecular area *A* were calculated according to Blecker et al., 2002 using Gibbs adsorption isotherm [52]:(2)$$\Gamma \;=\;\frac{1}{RT}\;\times \;(\frac{d\sigma }{\mathrm{dln}C}),$$
(3)$$A\;=\;\frac{1}{\Gamma \;\times \;N},$$
where *R* is the universal gas constant, *T* is the temperature in K, *σ* is the interfacial tension, *C* is the surfactant concentration, and *N* is the Avogadro number.

### 4.6. Dynamic Interfacial Tension and Interfacial Rheology Measurements

The dynamic interfacial tension and interfacial rheology of all solutions were determined using a pendant drop tensiometer (PAT1, Sintaface, Berlin, Germany). A drop of the respective solution with a surface area of 20 mm^2^ was produced from a cannula with 1 mm inner diameter. The interfacial tension was calculated from the drop shape over a period of 10,000 s maintaining a constant surface area. The surface area was then oscillatorically dilated for at least 10 oscillations with an amplitude of 2 mm^2^, followed by a 15-min oscillation pause at a constant surface area. The oscillation frequencies were 0.05, 0.1, 0.33, 0.5 and 0.67 Hz. The drop surface was dilated three times with oscillations of each frequency. The interfacial tension, interfacial viscosity, and interfacial elasticity were determined as described in Loglio et al. [53]. The respective mean value and deviation was calculated from two measurements with independently prepared solutions. The equilibrium interfacial tension and the time *t*_eq_, when equilibrium was reached, were taken when the interfacial tension changes became smaller than the deviation.

### 4.7. Foam Generation

A 50 mL VitaPor suction filter funnel (Por.4, 10–16 µm) from ROBU Glasgeräte GmbH (Hattert, Germany) was used for foam generation. 16 mL of surfactant solution (concentration = 2 × CMC) were filled into the suction filter funnel. Foam formation was initiated by introducing nitrogen with a gas flow of 60 mL/min through the funnel outlet. As soon as a foam height of 5.3 cm was reached, the nitrogen flow was stopped and the measurements were started.

### 4.8. Foam Height Measurements

Foam height was measured over a period of 60 min using a scale at the outside of the filter.

### 4.9. Bubble Size Distribution Measurements

Bubble sizes were analyzed using a VHX-950F microscope equipped with a VH-B55 endoscope both supplied by Keyence Deutschland GmbH (Neu-Isenburg, Germany). The endoscope, covered with a 90° angle mirror tube and inserted into a customized optical glass cuvette, was placed at a height of 22 mm above the filter. Pictures were taken every 15 s over a period of 10 min. For illumination the spotlight of a KL 1500 LCD goose neck lamp from Schott AG (Mainz, Germany) was placed at the outer wall of the filter funnel. The endoscopic pictures were evaluated using a software tool written in Matlab^®^ (MathWorks^®^, Natick, MA, USA) based on a template matching method as described by Völp et al. [54]. Bubble size distribution was analyzed in triplicates in freshly produced foams.

Coarsening rate *Ω* was obtained from the slope of square Sauter diameter versus time plots ($\Omega =\frac{{dD}^{2}}{dt}$) according to Briceño-Ahumada et al. [4]. Coarsening rates were calculated in a time range from 100–600 s, and correlation coefficients were at least 0.96.

### 4.10. Determination of Gas Volume Fractions

Conductivity was measured using a SevenCompact conductivity meter equipped with an Inlab^®^ 738 ISM four-electrode conductivity sensor from Mettler-Toledo (Schwerzenbach, Switzerland). The sensor was placed 22 mm above the filter membrane. The conductivity of the glycolipid solution was measured before foaming. The foam conductivity was measured every 15 s over a period of 10 min. The relative conductivity *κ_rel_* was calculated by (4):(4)$$\kappa_{\mathrm{rel}}\;=\;\frac{\kappa_{\mathrm{foam}}}{\kappa_{\mathrm{solution}}},$$
where *κ_foam_* is the conductivity of the foam and *κ_solution_* is the conductivity of the glycolipid solution.

The gas volume fraction *φ* was calculated as described by Feitosa et al. [55]:(5)$$\varphi \;=\;1\;-\;\frac{3\;\times \;\kappa_{\mathrm{rel}}\;\times \;(1+11\;\times \;\kappa_{\mathrm{rel}})}{{1\;+\;25\;\times \;\kappa_{\mathrm{rel}}+10\;\times \;\kappa_{\mathrm{rel}}}^{2}},$$

### 4.11. Determination of Shear Moduli

The shear modulus of the foams was determined using a RheoScope 1 rotational rheometer from Thermo Fischer Scientific (Karlsruhe, Germany) equipped with a plate-plate geometry with a diameter of 60 mm, covered with sandpaper (grit 40, average particle diameter 269 µm) to reduce wall slip effects. The gap height was set to 5 mm. A foam sample was prepared inside the filter funnel and approximately 20 mL of foam were transferred onto the bottom plate of the rheometer using a spoon 20 s before it reached the desired foam age. The device set the gap automatically within 20 s and the measurement started. The foams were sheared in oscillation with a fixed frequency of 1 Hz and the stress amplitude increased stepwise from 0.01 to 20 Pa in 12 logarithmically distributed steps during a measuring time of 60 s. The shear modulus was obtained from the average real part *G*’ of the shear modulus measured at stress amplitudes in the linear viscoelastic shear regime. In foams, the shear moduli are independent of the frequency typically in the range between 0.01 and 10 Hz [56] and since *G*’ is determined in this regime it is termed shear modulus *G*_0_.

### 4.12. Statistical Analysis

Results are given as mean ±standard deviation. Statistical data analysis was performed by two-way ANOVA and Tukey test using the OriginPro 9.6 (version 2019) software. Results were considered significant if *p*-value was <0.05.

## 5. Conclusions

The aim of this study was to investigate the structure–function relationship of seven enzymatically synthetized glycolipids with regard to their interfacial and foaming properties. Hereby, four different head groups, glucose, glucuronic acid, sorbose and sorbitol were evaluated, as well as unsaturation and branching in the C10 fatty acid tail.

Unsaturation in the fatty acid tail resulted in increased CMC and reduced interfacial elasticity, interfacial viscosity and foam stability. Branching also reduced interfacial elasticity and interfacial viscosity but increased foam stability. Glycolipids with different head groups showed only insignificant differences in interfacial rheological properties as well as foam elasticity. However, decylglucuronate and sorbose monodecanoate showed superior foam stability over glucose monodecanoate and sorbitol monodecanoate. These results indicate that among the tested sugar(-derivatives), ketoses and uronic acids have a higher potential as glycolipid head group for foaming applications than aldoses or alditols. Adsorption time at the interface was identified as crucial parameter for foam stability.

Consequently, this study reveals that both the head group, despite its minor influence on interfacial properties, and the functional groups in the fatty acid are crucial factors for foam stability. In a subsequent study, gas permeability, film thickness, film contact angle and surface forces of individual foam lamellae should be investigated in order to obtain more profound insights into the processes at the interfaces. Furthermore, technical characterization of the investigated glycolipids in terms of emulsification, greasing power and skin compatibility should be addressed.

## Figures and Tables

**Figure 1 molecules-25-03797-f001:**
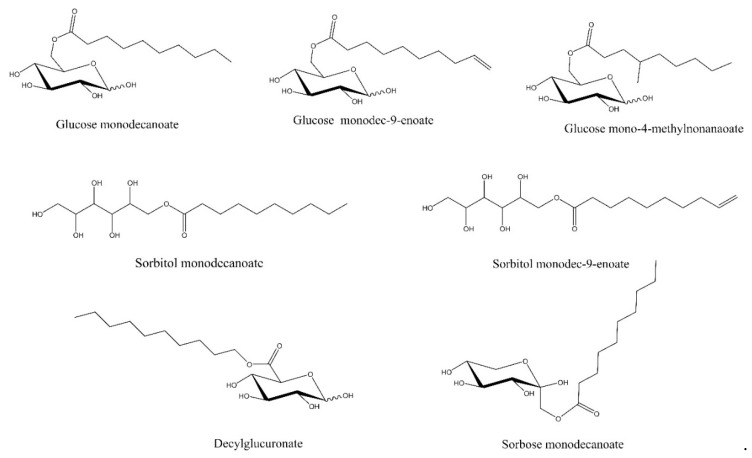
Structures of the investigated glycolipids.

**Figure 2 molecules-25-03797-f002:**
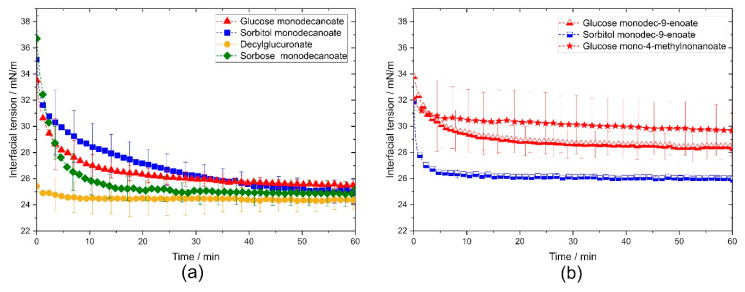
Dynamic interfacial tension of the investigated glycolipids vs. interface lifetime in pendant drop experiments. (**a**) Comparison of the different head groups. (**b**) Comparison of the different tail groups. Decylglucuronate reached equilibrium faster than glycolipids with glucose, sorbitol or sorbose head group. The branched glucose mono-4-methylnonanoate led to a faster reduction of interfacial tension than unbranched glycolipids.

**Figure 3 molecules-25-03797-f003:**
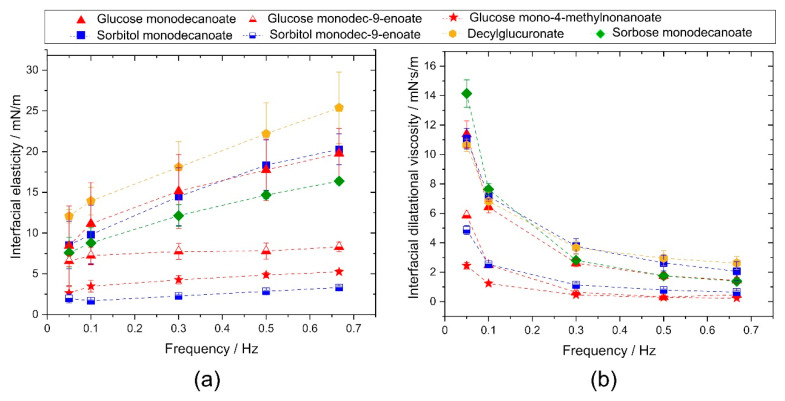
Interfacial rheological properties of the investigated glycolipids. (**a**) Complex interfacial dilatational elasticity modulus and (**b**) interfacial dilatational viscosity as a function of frequency. Unsaturated and branched glycolipids showed lower interfacial elasticity and interfacial viscosity than saturated, linear glycolipids.

**Figure 4 molecules-25-03797-f004:**
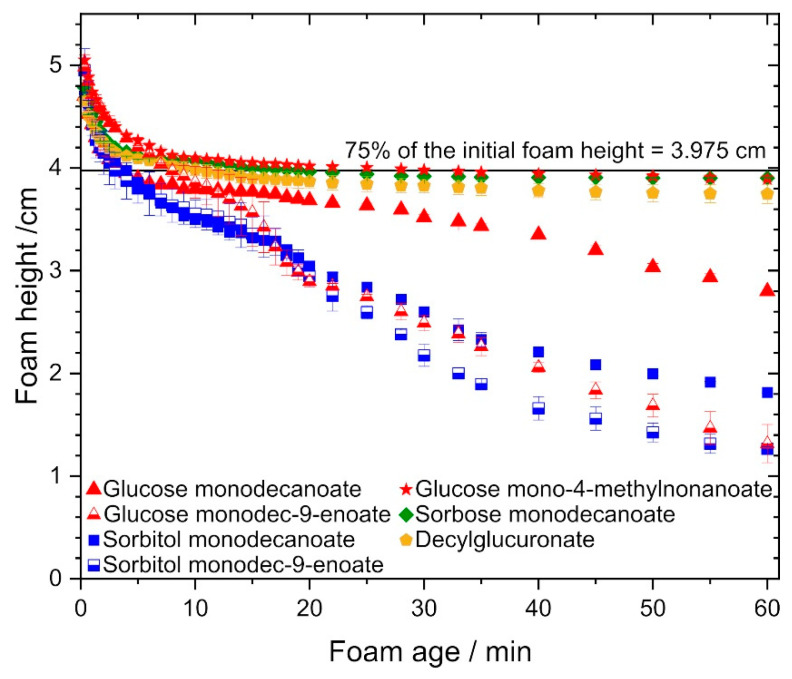
Foam stability of the investigated glycolipids, foam height vs. foam age. Foams stabilized by decylglucuronate, sorbose monodecanoate and glucose mono-4-methylnonanoate exhibited superior stability compared to the other glycolipids, while glycolipids with unsaturated fatty acid chain performed poorly.

**Figure 5 molecules-25-03797-f005:**
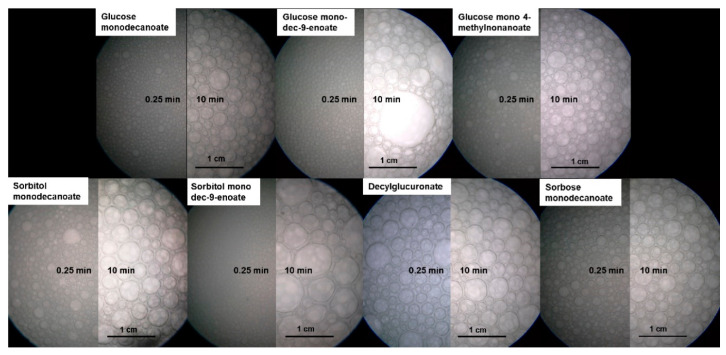
Endoscopic pictures of foams stabilized by glycolipids at 15 s and 600 s foam age. In foams stabilized by unsaturated glycolipids some huge bubbles occurred while bubble size of the other glycolipids is more homogenous.

**Figure 6 molecules-25-03797-f006:**
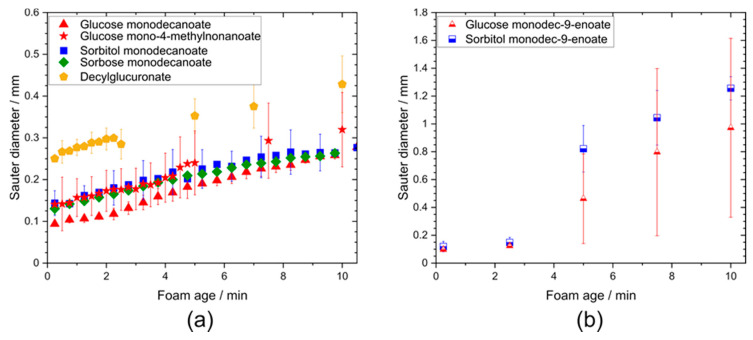
Sauter diameter of foams stabilized by different glycolipids as a function of foam age. (**a**) Comparison of different head groups and influence of branching in the tail group. (**b**) Impact of unsaturation in the hydrophobic tail group on Sauter diameter. Sauter diameter of foams stabilized by unsaturated glycolipids raised faster than those of the other glycolipids.

**Figure 7 molecules-25-03797-f007:**
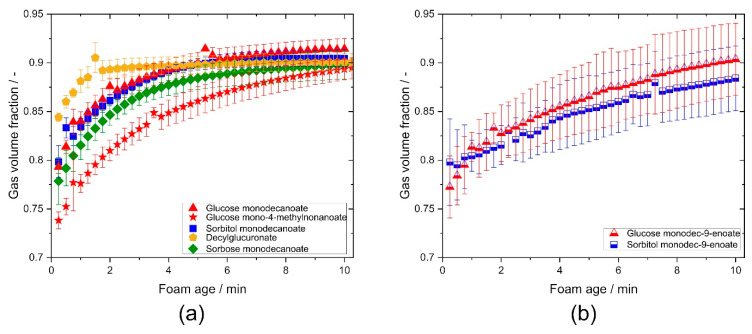
Gas volume fraction of the investigated glycolipids vs. foam age. (**a**) Comparison of glycolipids with different head groups and influence of branching in the fatty acid tail. (**b**) Comparison of saturated and unsaturated glycolipids.

**Figure 8 molecules-25-03797-f008:**
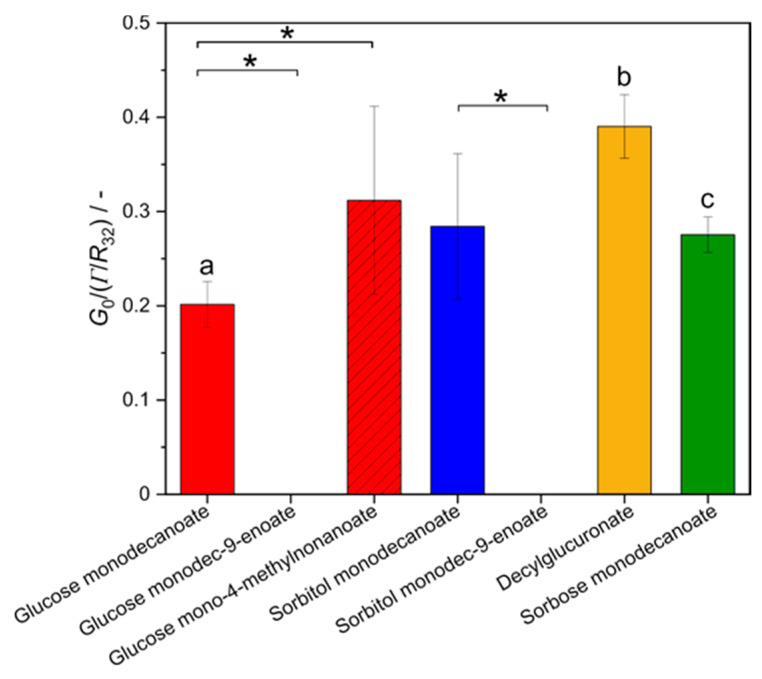
Shear modulus G_0_ of glycolipid foams normalized by Laplace pressure *Γ*/*R*_32_ at a gas volume fraction of 0.88. * indicates statistically significant differences between tail groups. The letters a, b, and c indicate statistically significant differences between head groups. Unsaturation and branching in the hydrophobic fatty acid tail significantly influence the shear modulus G_0_ while different head groups affect the shear modulus G_0_ only slightly.

**Figure 9 molecules-25-03797-f009:**
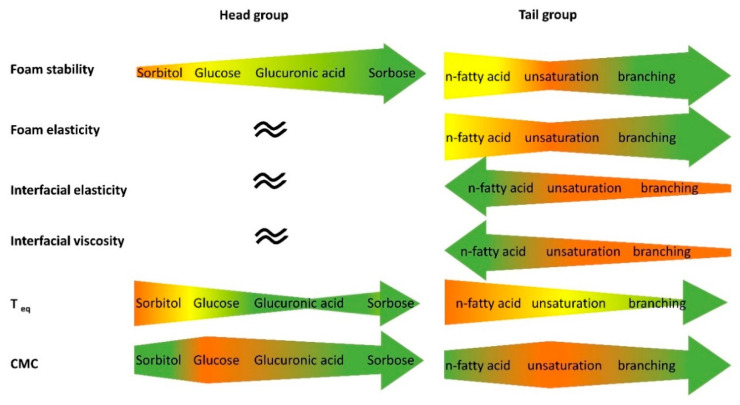
Summary of the results of the interfacial and foaming investigations. The width of the arrow indicates the size of the respective parameter. Green indicates higher interfacial activity/foam stabilizing property/elasticity. *t*_eq_ is the time when equilibrium interfacial tension is reached.

**Figure 10 molecules-25-03797-f010:**
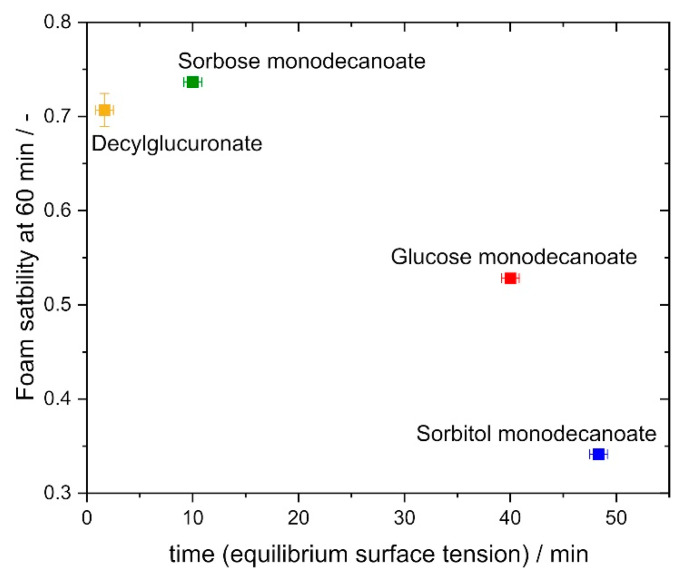
Correlation of foam stability at 60 min with time to reach equilibrium surface tension. Foams stabilized by glycolipids with a faster adsorption at the interface exhibit higher foam stability. $\mathrm{Foam}\;\mathrm{stability}\;=\;\frac{\mathrm{Foam}\;\mathrm{height}\;\mathrm{at}\;60\;\min}{\mathrm{Initial}\;\mathrm{foam}\;\mathrm{height}}$.

**Table 1 molecules-25-03797-t001:** Interfacial properties of the investigated glycolipids, including critical micelle concentration (CMC), equilibrium interfacial tension and molecular area.

Glycolipids	CMC in mM	Interfacial Tension in mN/m ^1^	*t*_eq_ in s	Molecular Area in Å^2^/Molecule
Glucose monodecanoate	1.5	25.5 ± 0.17 ^a^	2400	26.3
Glucose monodec-9-enoate	3.0	28.5 ± 1.10 ^b,c^	1100	30.7
Glucose mono-4-methylnonanoate	1.8	29.6 ± 1.90 ^b^	300	39.4
Sorbitol monodecanoate	0.7	24.9 ± 0.84 ^a^	2900	32.6
Sorbitol monodec-9-enoate	3.0	26.0 ± 0.13 ^a,c^	700	42.1
Decylglucuronate	1.3	24.3 ± 0.63 ^a^	100	33.7
Sorbose monodecanoate	1.0	25.0 ± 1.10 ^a^	600	30.9

^1^ (c = 2 × CMC). ^a,b^ indicate statistical significant differences; ^c^ indicates statistical significant difference to the corresponding glycolipid with saturated fatty acid tail. *t*_eq_ is the time when equilibrium interfacial tension is reached.

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
