# Peer review of "Interfacial and Foaming Properties of Tailor-Made Glycolipids—Influence of the Hydrophilic Head Group and Functional Groups in the Hydrophobic Tail"

_molecules, 2020, doi:10.3390/molecules25173797_

Round 1
Reviewer 1 Report
I found this manuscript as an interesting and well written study on new and green surfactants. In my opinion, presented topic is very important and catchy nowadays. The laboratory work was well done and a lot of data on synthesized glycolipids were delivered. Of course, there are few minor issues (listed below) that might be reconsidered by Authors before final version is presented. My recommendation is: publish after minor revision.
List of issues:
- Authors should do careful proof read to avoid issues as: ‘…For foams of from glycolipids…’ (page 9 line 262)
- Is it possible to move ‘Materials and Methods’ before ‘Results’? It is slightly unusual for me if ‘Discussion’ is not followed by ‘Conclusions’ directly.
- Figure 6B should present ‘impact of unsaturation in the hydrophobic tail group on Sauter diameter’, but graphs’ legend suggest something else.
- Page 11, line 340: Figure A2c is mentioned but there is no such a figure in the appendix.
- ‘2.1. Critical micelle concentration (CMC) and dynamic interfacial tension’ – it is interesting to see figures presenting kinetics of surfactant adsorption, but I think that for a potential reader it might be useful to see an adsorption isotherm as well. Also it might be interesting to fit some theoretical model to the experimental data (Langmuir or Frumkin isotherm). In my opinion this is more proper way to obtain a molecular area (see Mobius et al. Surfactants: Chemistry, Interfacial Properties, Applications, 2001).
Author Response
Dear Reviewer 1,
thank you for your insightful comments that helped to improve the manuscript.
Manuscript was carefully checked for spelling and grammar issues like in line 262.
We used the article template of the journal. The order of the chapters was not adjusted by us.
The name of the mentioned figure was adjusted. We are sorry for that typo.
The surface excess and surface area data shown in this study were obtained by Gibbs adsorption isotherm which is widely used in studies on glycolipids (Blecker et al. 2002, Garofalakis et al. 2000, ...). This was stated in the materials section more clearly now. Gibbs adsorption isotherm and Langmuir isotherm are reported to give comprable data (Martinez-Balbuena et al 2017 Applicability of the Gibbs Adsorption Isotherm to the analysis of experimental
surface-tension data for ionic and nonionic surfactants; Zdzienka et al 2017 Thermodynamic parameters of some biosurfactants and surfactants
adsorption at water-air interface). Therefore, we don't think that it has benefits for the manuscript to calculate surface excess additionally by another theoretical model which is known to result in comparable results as the one we used.
Best regards
Rebecca Hollenbach
Reviewer 2 Report
The authors studied the interfacial and foaming properties of seven enzymatically synthesized glycolipid surfactants with different hydrophilic and hydrophobic groups. The hydrophilic groups used included glucose, sorbitol, glucuronic acid and sorbose, whereas the hydrophobic domains included saturated, unsaturated and branched acyl chains (decanoate, dec-9-enoate and 4-methyl-nonanoate). Surface activity and rheological properties were studied, including equilibrium interfacial tension, dynamic interfacial tension, foam formation, foam stability and bubble size distribution.
Glycolipids with a saturated fatty acid exhibited better surface activity (lower critical micelle concentration and lower interfacial tension values) when compared with those containing unsaturated or branched moieties. However, an opposite behavior was observed regarding the adsorption time required to achieve the equilibrium interfacial tension. The same differences were observed for their rheological properties. Regarding foam stability, the best results were obtained for glycolipids containing branched fatty acids, followed by saturated fatty acids.
The results are properly presented and organized, with clear figures and tables. Furthermore, they are properly discussed. All the conclusions are supported by the data presented by the authors. The results obtained are relevant from an industrial point of view, as they provide relevant data for the appropriate design of surface active compounds with better foam stabilizing properties.
There is a mistake in the legend of Figure 7b: Glucose monodecanoate and sorbitol monodecanoate should be glucose mono-dec-9-enoate and sorbitol mono-dec-9-enoate, respectively.
According to lines 230-234, there are significant differences in the normalized shear modulus between foams stabilized with glucose monodecanoate and glucose mono-4-methyldecanoate. However, there are no significant differences between decylglucuronate and glucose monodecanoate or sorbose monodecanoate. Looking at Figure 8, it does not seem like that.
Line 250-254: That behavior was also reported for rhamnolipdis (Zhang, L., Pemberton, J.E., Maier, R.M., 2014. Effect of fatty acid substrate chain length on Pseudomonas aeruginosa ATCC 9027 monorhamnolipid yield and congener distribution. Process Biochem. 49, 989–995).
Author Response
Dear Reviewer 2,
thank you for your insightful comments that helped to improve the manuscript.
The mistake in the legend of figure 6B was corrected.
The statistical analysis of the foam elasticity was carefully checked and lines 230-234 as well as Figure 8 were adjusted.
Line 250-254: it was added that this effect is also reported for rhamnolipids.
Best regards
Rebecca Hollenbach